# CONTEXT AND HISTORY AWARE OTHER-SHAPING

## ABSTRACT

Cooperation failures, in which self-interested agents converge to collectively worst-case outcomes, are a common failure mode of Multi-Agent Reinforcement Learning methods. Methods such as Model-Free Opponent Shaping and The Good Shepherd address this issue by shaping their co-player's learning into mutual cooperation. However, these methods fail to capture important co-player learning dynamics or do not scale well to co-players parameterised by deep neural networks. To address these issues, we propose *Context and History Aware Other-Shaping* (CHAOS). A CHAOS agent is a meta-learner that learns to shape its co-player over multiple trials. CHAOS considers both the *context* (inter-episode information), and *history* (intra-episode information) to shape co-players. CHAOS also successfully scales to shaping co-players parameterised by deep neural networks. In a set of experiments, we show that CHAOS achieves state-of-the-art shaping in matrix games. We provide extensive ablations, motivating the importance of both context and history. CHAOS also successfully shapes on a grid-world-based game, demonstrating CHAOS' scalability empirically. Finally, we provide empirical evidence that, counterintuitively, the widely-used Coin Game environment does not require history to learn shaping because states are often indicative of past actions, making it unsuitable for investigating shaping.

## 1 INTRODUCTION

Multi-agent learning has shown great success in strictly competitive (Silver et al., 2016) and fully cooperative settings (Foerster et al., 2019; Rashid et al., 2018). In competitive games, agents can learn Nash equilibrium strategies by iteratively best-responding to suitable mixtures of past opponents. Similarly, best-responding to rational co-players leads to the desirable equilibria in cooperative games (assuming joint training). In contrast, Nash equilibria often coincide with globally worst welfare outcomes in general-sum games, rendering the aforementioned learning paradigms ineffective. For example, in the iterated prisoner's dilemma (IPD) (Axelrod & Hamilton, 1981; Harper et al., 2017), naive best-response dynamics converge on unconditional mutual defection (Foerster et al., 2018).

The above methods ignore a crucial factor: when multiple learning agents interact in a shared environment, the actions of one agent influence the environment and, often, the reward of other agents, which in turn influences their *learning dynamics*. For example, a car merging into the middle lane in heavy traffic makes it unattractive for fellow collision-averse motorists to move to the middle lane at the same time. Our paper investigates methods which allow agents to exploit this interconnection between their actions and the learning outcome of other agents and leverage it to their advantage. Such "shaping" methods explicitly account for other agent's learning and have achieved promising results, e.g. discovering the prosocial tit-for-tat strategy in the IPD (Foerster et al., 2018; Letcher et al., 2019b; Willi et al., 2022; Balaguer et al., 2022; Lu et al., 2022). However, early shaping methods are myopic (only shape the next learning step of the co-player), require white-box access to the co-player's parameters and require higher-order derivatives. To overcome these shortcomings, both Model-Free Opponent Shaping (Lu et al., 2022, M-FOS) and The Good Shepherd (Balaguer et al., 2022, GS) frame shaping as a meta reinforcement learning problem. In these approaches, the meta-agent learns to shape others by observing full training runs of the co-players in each meta-training episode before updating its policy. M-FOS and GS showed promising empirical success. However, both methods have shortcomings: M-FOS's meta-agent outputs a policy parameterisation for the inner-agent (similar to HyperNetworks (Ha et al., 2017)). This limits M-FOS to games where the

policies can be represented compactly, such as in *infinitely-iterated* matrix games. While M-FOS reports results in a higher-dimensional game (in which neural networks represent the policies), it uses a hierarchical architecture. GS does not output whole parameterisations but instead keeps its policy fixed during the entire duration of a trial. This prevents GS from using the training *context* to shape the co-player adaptively.

To address both issues, we propose Context and History Aware Other-Shaping[1] (CHAOS). In CHAOS, the meta-agent and the inner agent it controls are parameterised by a single recurrent neural network (RNN). A CHAOS agent meta-learns by retaining its hidden state throughout an entire meta-episode, similar to RL$^2$ (Duan et al., 2016) in single-agent RL. This hidden state enables CHAOS agents to react to two components of the co-player's learning: The *context* - inter-episode learning and the *history* - intra-episode behaviour. In shaping problems, *history* captures the co-player's current policies whilst context captures the co-player's learning rules. Together these two enable CHAOS to dynamically shape agents. Combining the meta-agent and the inner agent into one recurrent meta-learner avoids outputting policy parameterisations, unlike M-FOS.

We show that CHAOS discovers a ZD-extortion-like strategy in the *finitely-iterated* prisoner's dilemma (a more challenging setting than the *infinitely-iterated* PD where the environment is non-differentiable and where policies cannot be represented compactly). Moreover, we show that CHAOS matches or outperforms GS and M-FOS in iterated matrix games. CHAOS also matches state-of-the-art shaping against memory-based agents in the Coin Game, a grid-based environment where deep neural networks represent policies.

To summarise our contributions

- We introduce CHAOS, a shaping method capturing both learning context and history, suitable for high-dimensional games.
- We formalise the concept of history and context for shaping and analyse their respective roles empirically.
- We demonstrate state-of-the-art performance on a set of iterated matrix games.
- We identify a fundamental problem in the widely-used Coin Game.

## 2 RELATED WORK

**Opponent Shaping** Many methods exist that explicitly account for their opponent's learning. Just like CHAOS, these approaches recognise that the actions of any one agent influence their co-players policy and seek to use this mechanism to their advantage (Foerster et al., 2018; Letcher et al., 2019a; Kim et al., 2021a; Willi et al., 2022). However, in contrast to CHAOS, these approaches require privileged information to shape their opponents. These models are also myopic since anticipating many steps is intractable. Balaguer et al. (2022) and Lu et al. (2022) solve the issues above by framing opponent shaping as a meta reinforcement learning problem, which CHAOS inherits and builds upon. The specific differences to M-FOS and GS will be the subject of Section 4.

**Opponent Modelling** Similarly to our work, opponent modelling tries to disentangle some aspects of other agents' policies from the environment (Mealing & Shapiro, 2017; Raileanu et al., 2018; Tacchetti et al., 2018). In contrast to our work, these approaches do not consider agents as learners. Furthermore, they do not observe agents at different stages of learning and thus, whilst modelling as non-stationary, do not observe learning dynamics (Synnaeve & Bessière, 2011). Finally, CHAOS does not *explicitly* model any aspect of the opponent.

**Multi-Agent Meta-Learning** Multi-agent meta-learning approaches have also shown success in mixed-games with other learners (Al-Shedivat et al., 2018; Kim et al., 2021b; Wu et al., 2021). Similar to CHAOS, they take inspiration from meta reinforcement learning - their approach is to learn the optimal initial parameterisation for the shaper akin to Model-Agnostic Meta Learning (Finn et al., 2017). In contrast, CHAOS uses an approach similar to RL$^2$ (Duan et al., 2016), which trains an RNN-based agent to implement efficient learning for its next task. Furthermore, CHAOS is optimised using evolution strategies (Salimans et al., 2017), allowing it to consider much longer time horizons than policy-gradient metods (Schulman et al., 2017).

---

[1]"Other" breaks with the line of seminal work on *opponent shaping*, but highlights the general-sum aspect

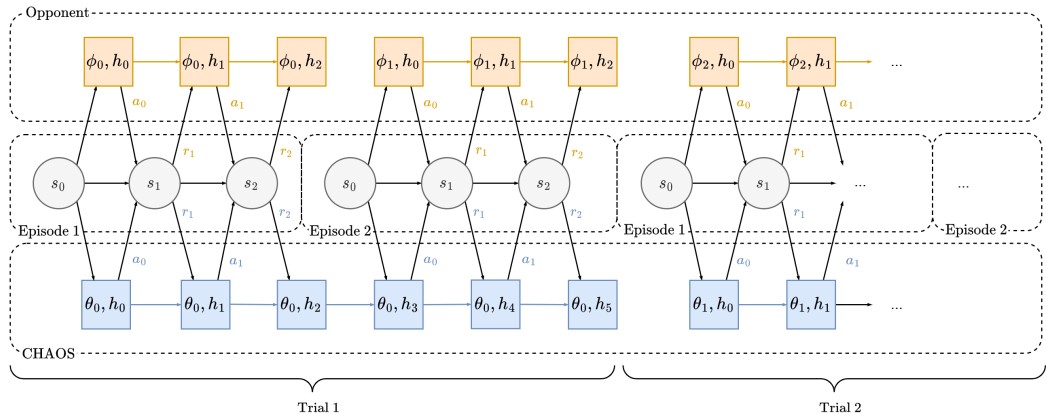

Figure 1: Example meta-learning interaction. CHAOS $\theta$ (blue) resets its hidden state $h$ at the beginning of a trial, updates its hidden state after each environment interaction, then updates its parameters at the end of a trial. The opponent $\phi$ (orange) resets its parameters at the beginning of a trial, then updates its parameters at the end of each episode. Optionally, if the opponent uses memory, its hidden state $h$ is reset at the beginning of a trial and updated after each environment interaction.

## 3 BACKGROUND

**Partially Observable Stochastic Game (POSG)** A POSG is given by the tuple $\mathcal{M} = \langle \mathcal{N}, \mathcal{A}, \mathcal{O}, S, \mathcal{T}, \mathcal{I}, \mathcal{R}, \gamma \rangle$, where $\mathcal{A}$, $\mathcal{O}$, and $S$ denote the action, observation, and state space, respectively. These parameters can be distinct at every time step and also incorporated into the transition function $\mathcal{T} : S \times \mathbf{A} \to \Delta S$, where $\mathbf{A} \equiv \mathcal{A}^n$ is the joint action of all agents. Each agent draws individual observations according to the observation function $\mathcal{I} : S \times N \to \mathcal{O}$ and obtains a reward according to their reward function $\mathcal{R} : S \times \mathbf{A} \times N \to \mathbb{R}$ where $N = \{1, \ldots, n\}$. POSGs can represent general-sum games. The single player case, $N = \{1\}$, of POSGs are Partially Observable Markov Decision Processes (POMDPs).

**Evolution Strategies (ES)** CHAOS uses Evolution Strategies (Salimans et al., 2017, ES) to optimise the meta-agent. ES is a model-free optimisation method. Let $F : \mathbb{R}^d \to \mathbb{R}$ be some function we want to optimise over. Instead of optimising the objective directly, ES blurs $F(\mathbf{x})$ with Gaussian noise

$$\mathbb{E}_{\epsilon \sim N(\mathbf{0}, I_d)}[F(\mathbf{x} + \sigma\epsilon)],$$

where $\sigma$ is a hyper-parameter controlling how much Gaussian noise is added. This allows using the following simple gradient estimator:

$$\nabla_{\mathbf{x}} \mathbb{E}_{\epsilon \sim N(\mathbf{0}, I_d)}[F(\mathbf{x} + \sigma\epsilon)] = \mathbb{E}_{\epsilon \sim N(\mathbf{0}, I_d)}\left[\frac{\epsilon}{\sigma} F(\mathbf{x} + \sigma\epsilon)\right],$$

allowing the optimization of non-differentiable functions using gradient descent methods. ES bypasses the credit assignment problem by directly optimizing in the parameter space of the model instead of the policy space and is thus suitable for long-time horizon problems (Salimans et al., 2017).

**RL$^2$** CHAOS takes an RL$^2$-like approach to meta-learning. RL$^2$ is a single-agent meta-reinforcement learning method where the meta-agent is parameterised by a recurrent neural network $(\phi, h)$, where $\phi$ are the network parameters and $h$ the hidden state. RL$^2$ samples MDPs from a distribution $\rho_{\mathcal{P}} : \mathcal{P} \to \mathbb{R}_+$ and interacts with each MDP sequentially over a *trial*, consisting of a number of episodes E. Importantly, the RL$^2$ agent retains its hidden state $h$ across all episodes in a trial and resets only when it faces a new trial, i.e. a new MDP is sampled. The objective is to maximise the expected total discounted reward over a trial $t$ instead of a single episode

$$\mathbb{E}_{\phi^t}\left[\sum_{l=0}^{L} \gamma^l r\left(s_l, a_l\right)\right]$$

where $L = K * E$ and $K$ is the episode length. The RL$^2$ agent is thus encouraged to use all the information captured in its hidden state $h$. RL$^2$ agents have been shown to scale to high-dimensional problem settings (Duan et al., 2016).

Table 1: Converged Reward $(row, column)$ for agents against Naive Learners on the Iterated Prisoners Dilemma (IPD), Iterated Matching Pennies (IMP) and Coin Game, reported across 5 seeds with standard deviations. Reward is averaged per (step/episode) for (Matrix/Coin) game respectively.

| | IPD | IMP | Coin Game |
|---|---|---|---|
| CHAOS | **-0.13 ± 0.02, -2.84 ± 0.05** | **0.86 ± 0.02, -0.86 ± 0.02** | $6.51 \pm 0.46, 2.71 \pm 0.11$ |
| M-FOS | -0.60 ± 0.14, -2.34 ± 0.14 | 0.83 ± 0.09, -0.83 ± 0.09 | $2.67 \pm 0.52, 3.94 \pm 0.15$ |
| GS | -0.97 ± 0.03, -1.26 ± 0.10 | -0.01 ± 0.01, 0.01 ± 0.01 | **6.72 ± 0.72, 2.39 ± 0.10** |
| NL | -2.00 ± 0.00, -2.00 ± 0.00 | 0.00 ± 0.00, 0.00 ± 0.00 | $0.47 \pm 0.83, 0.26 \pm 0.30$ |

(a)  (b)  (c)

Figure 2: Evaluation results in the finite IPD over a single trial composed of 100 inner episodes over 20 seeds, where (a) shows the reward, (b) CHAOS' conditional probability of cooperation, and (c) state visitation.

**Good Shepherd** GS formalises shaping as a meta-learning problem over a sequence of trials of length $T$. Each trial contains $E$ episodes. In each trial, GS shapes a new co-player in a POSG $\mathcal{M}$, where $(\phi_i, \phi_{-i})$ correspond to GS' and the co-players' parameters, respectively. During a trial $t$, GS uses a fixed policy $\phi_i^t$. At the end of each inner episode, the co-players update their parameters with respect to the episodic return $J_{-i}^e = \sum_{k=0}^K r_{-i}^k(\phi_i^t, \phi_{-i}^e)$, where $K$ is the length of an episode. For example, if the co-players were Naive Learners, the update looks as follows: GS optimises the meta-return $\bar{J} = \sum_e^E J_i^e$ (summed over all episodes) at the end of a trial using Evolution Strategies (Salimans et al., 2017). The policy is parameterised by a feed-forward network, thus lacking both history and context. Therefore, GS cannot adapt to changing learning dynamics of the co-player, which is detrimental, as we show in Section 6.

**Model-Free Opponent Shaping** M-FOS frames opponent shaping as a meta reinforcement task. More specifically, the meta-task is formulated as a POMDP $\langle \overline{\mathcal{S}}, \overline{\mathcal{A}}, \Omega, \overline{\mathcal{O}}, \overline{\mathcal{P}}, \overline{\mathcal{R}}, \bar{\gamma} \rangle$ over an underlying general-sum game, represented by a POSG $\mathcal{M}$. In the POMDP, the meta-state $\overline{\mathcal{S}}$ spans the policies of every player in the underlying POSG: $\bar{s}_e = (\phi_i^{e-1}, \phi_{-i}^{e-1}) \in \mathcal{S}$. The meta-action space $\overline{\mathcal{A}}$ consists of the policy parameterisation of the underlying inner agent playing the game for the meta-agent. At each meta-episode, conditioned on both agent's policies, M-FOS outputs parameters of the next inner-agents (similar to a HyperNetwork (Ha et al., 2017)), i.e., $\bar{a}_e = \phi_i^e \sim \pi_\theta (\cdot \mid \bar{o}_e)$, where $\theta$ is the parameters of the M-FOS agent. The meta-reward is the return of the inner agent over one inner episode, $\bar{r}_e = \sum_{k=0}^K r_i^k (\phi_i^e, \phi_{-i}^e)$. This Hypernetwork-like approach is M-FOS' main shortcoming - it is difficult to scale to complex inner-agent policy parameterisations. To scale to complex inner-agent policy parameterisations that are used in the Coin Game environment, they use a hierarchical architecture in which the meta-agent instead outputs a *conditioning vector* for the inner agent that contains context information (Lu et al., 2022). M-FOS is optimised using Evolution Strategies (Salimans et al., 2017) for the iterated matrix games and uses PPO (Schulman et al., 2017) for the Coin Game environment.

## 4 METHOD

We assume a set of POSGs $M$ and a distribution we can sample from $\rho_M$. We also assume a set of co-player initialisations $\phi_i^0$ and a corresponding distribution $\rho_\phi$, as shaping acknowledges other learners within the environment, in contrast to RL$^2$. Just like in GS, we define a *inner episode* to be a finite sequence of interactions within a fixed POSG $\mathcal{M} = \langle \mathcal{N}, \mathcal{A}, \mathcal{O}, S, \mathcal{T}, \mathcal{I}, \mathcal{R}, \gamma \rangle$ and fixed initial learner, and a *trial* to be a sequence of inner episodes.

Figure 1 illustrates the interaction between agents and the environment. First, the shaping agent's parameters $\phi_i$ and hidden state $h_i$ are initialised. At the start of a trial, co-players $\phi_{-i} \sim \rho_\phi$ and a new game (POSG) $\rho_\mathcal{M}$ are sampled. During an episode of length $K$, upon receiving a state, agents take their respective actions, $a_i^k \sim \pi_{\phi_i}(\cdot \mid o_i^k, h_i^t)$. At each time step in the episode, the internal state of the shaping agent is updated: $h_i^{k+1} = f(o_i^k, h_i^k)$. On receiving actions, the POSG returns rewards $r_i^k$, new observations $o_i^{k+1}$ and a done flag $d$, indicating if an episode has ended.

When an inner episode terminates, the co-player takes a gradient update maximising total *episode* return, $J_{-i}^e = \sum_{k=0}^K r_{-i}^k(\phi_i^t, \phi_{-i}^e)$. The updated co-player $\phi_{-i}^{e+1}$ and the shaper's hidden state $h_i^K$ are passed to the next episode. This process is repeated over $E$ episodes in a trial. When a trial terminates, the shaper's policy is updated, maximising total *trial* reward, $\bar{J} = \sum_e^E J_i^e$, via ES. This leads to the following objective,

$$\max_{\phi_i} \mathbb{E}_{\rho(\phi),\rho(\mathcal{M})} \left[ \bar{J} \right].$$

**Algorithm 1:** CHAOS update algorithm. Given a distribution of game functions $M \sim \rho_M$, returns $J_i, J_{-i}$, policies $\pi_{\phi_i}, \pi_{\phi_{-i}}$ and their respective initial hidden states $h_i, h_{-i}$ and a distribution of initial co-players $\rho_\phi$, this algorithm updates a shaper policy $\phi_i$ over $T$ trials consisting of $E$ episodes.

**Require:** $\phi_i, \rho_\phi, \rho_M, E, T$
1: **for** $t = 0$ **to** $T$ **do**
2:    Initialise trial reward $\bar{J} = 0$
3:    Initialise shaper hidden state $h_i = \mathbf{0}$
4:    Sample co-players $\phi_{-i} \sim \rho_\phi$
5:    **for** $e = 0$ **to** $E$ **do**
6:       Sample game $\mathcal{M} \sim \rho_\mathcal{M}$
7:       Initialise co-players' $h_{-i} = \mathbf{0}$
8:       $J_i, J_{-i}, h_i', h_{-i}' = \mathcal{M}(\phi_i, \phi_{-i}, h_i, h_{-i})$
9:       Update co-players'
         $\phi_{-i} \leftarrow \phi_{-i} + \nabla_{\phi_{-i}} J_{-i}$
10:      $h_i \leftarrow h_i'$
11:      $\bar{J} \leftarrow \bar{J} + J_i$
12:    **end for**
13:    Update $\phi_i$ with respect to $\bar{J}$
14: **end for**

We use Evolution Strategies (Salimans et al., 2017) to maximise the expected fitness across a population. At the start of each generation, a population of CHAOS agents of size $M$ and $N$ Naive Learners (NL) are initialised. During a generation, the population plays against copies of the Naive Learner in parallel for a series of $E$ inner episodes. At the end of an inner episode, each copy of the Naive Learner performs a gradient update. At the end of the trial, CHAOS performs a gradient update in the direction of the maximum expected fitness, and the next generation begins.

**Difference to M-FOS and GS** CHAOS can be understood as a simpler version of M-FOS in which the meta-agent and underlying agent are collapsed into a single agent. To contrast CHAOS to M-FOS in more detail, we differentiate between the method definition and the actual architecture.

In the method definition of M-FOS, there are two action spaces: the meta-action space $\bar{\mathcal{A}}$ and the underlying action space $\mathcal{A}$. In CHAOS, the only action space is that of the underlying game, meaning *there is only one agent in CHAOS, whereas there are two agents in M-FOS*. Consequently, CHAOS can be understood as a special case of M-FOS. To make this explicit, let us define history as trajectory of an episode $e$, $\tau_e = \left(o_e^0, a_e^0, r_e^0, ..., r_e^K\right)$, and context as a trajectory of a trial t, $\bar{\tau}_t = (\tau_0, ..., \tau_E)$. Thus we can express the policies of CHAOS, M-FOS and GS as the following:

$$
\begin{aligned}
&\text{CHAOS:} &&a \sim \pi_\phi(\cdot \mid s, \bar{\tau}_t, \tau_e) \\
&\text{M-FOS:} &&a \sim \pi_{\phi \sim \pi_\theta(\cdot \mid \bar{\tau}_t)}(\cdot \mid s, \tau_e) \\
&\text{GS:} &&a \sim \pi_\phi(\cdot \mid s, \tau_e)
\end{aligned}
\tag{1}
$$

Thus it becomes apparent that 1) M-FOS' use of an outer agent is redundant and 2) that GS is unable to condition upon the context.

Secondly, CHAOS' architecture is different from the architectures proposed in M-FOS. M-FOS proposes *two* significantly different architectures (Lu et al., 2022). For matrix games, where a table can represent the policy, the meta-agent is a feedforward neural network whose output is the exact policy of the underlying agent. In this case, the meta-agent is trained using Evolutionary Strategies. For the Coin Game, in which a simple table cannot represent policies, M-FOS proposes an architecture akin to Hierarchical RL. Both the meta-agent and the underlying agent are recurrent neural networks. The underlying agent resets their hidden state after each episode, whereas the meta-agent does not. In this architecture, the meta-agent does not output the full parameterisation of the underlying agent

but instead outputs a conditioning variable, which the underlying agent uses as input. M-FOS relies on the assumption that outputting a conditioning variable is equivalent to outputting a policy parameterisation. The conditioning variable is fixed during an episode. In contrast, CHAOS only uses one recurrent neural network that does not reset its hidden state after an episode. *Both M-FOS and CHAOS can capture context and history; however, CHAOS only needs one agent to do so.*

Next, we compare CHAOS to GS. Originally, GS does not use a recurrent agent, and there is no discussion about notions of history or context. Thus GS cannot capture history or context. Even though it is not discussed, the framework is extensible to a recurrent meta-agent. However, that meta-agent only captures history as the hidden state is reset after each episode. By not capturing context, GS fails to shape in zero-sum games, such as the Matching Pennies, which we show in our experiments.

## 5 EXPERIMENTS

### 5.1 ENVIRONMENTS

**Iterated Prisoner's Dilemma** is a well-known and widely studied general-sum game illustrating that two rational agents may not cooperate even if it is globally optimal. The players choose to either cooperate (C) or defect (D) and receive a payoff according to Table 2a. In the *iterated* prisoner's dilemma (IPD), the agents repeatedly play the prisoner's dilemma and can observe the previous decisions.

Past research has used the infinitely IPD in their experiments (Foerster et al., 2018; Letcher et al., 2019b; Willi et al., 2022; Lu et al., 2022; Balaguer et al., 2022). In the infinite version, players submit a policy represented by five parameters, where each parameter is the probability of cooperation after each state given a one-step history ($CC, CD, DC, DD$, start). Press & Dyson (2012) showed that having access to the last state is sufficient for acting optimally. The infinite version is a differentiable game Balduzzi et al. (2018) as the exact value function can be calculated directly from the policies, thus accessing exact gradients is possible and optimization tractable. In our work, we consider the finitely IPD (f-IPD), and we do not take advantage of exact value functions, resulting in a version of the game that is more similar to current reinforcement learning environments. In the f-IPD, the agents do not submit a full strategy but take an individual action (either C or D) at each timestep.

Over repeated interactions, IPD produces a spectrum of interesting behaviours. In particular, two cases are of interest in this work: 1) Cooperation, and 2) Zero-Determinant (ZD) Extortion strategies. In Cooperation, agents are shaped sufficiently to CC and choose not defect, even though this would increase short-term rewards. In ZD-Extortion (Press & Dyson, 2012), co-players cooperate while allowing the shaper to enforce a linear relationship between their own payoff and that of the co-player, thus inducing behaviours that are more favourable than mutual cooperation.

**Iterated Matching Pennies** (IMP) is an iterated matrix game like the IPD. The players choose either heads (H) or tails (T), and receive a payoff according to the choices of both players. In contrast to the IPD, which is a general-sum game, IMP is a zero-sum game. The only equilibrium strategy for each one-memory agent is to play a random policy, resulting in an expected joint payoff of (0,0). It is only with intra-episode memory that a shaper can observe a co-player's policy and begin shaping.

**Coin Game** is a multi-agent, wrap-around grid-world environment that simulates social dilemmas (like the IPD) with high-dimensional states and multi-step actions (Lerer & Peysakhovich, 2017). Two players – blue and orange – move around a grid and pick up blue and orange coloured coins. When a player picks up a coin of its own colour, the player receives a reward of $+1$. When a player picks up a coin of the co-player's colour, the player also receives a reward of $+1$ and the co-player receives a reward of $-2$. If a coin gets picked up, a new coin of the same colour appears in a random location on the grid. If both agents reach a coin simultaneously, then both agents pick up that coin (the coin is duplicated). The episodes are of fixed length. When both players pick up coins without regard to colour, the expected reward is $0$. In contrast to the IPD, the Coin Game requires learning from high-dimensional states, a task that current shaping methods struggle to learn.

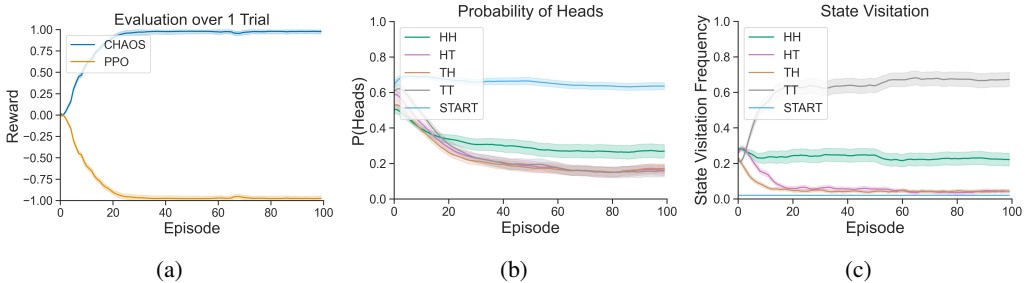

Figure 3: Evaluation results over a single trial (with novel co-player) composed of 100 inner episodes over 20 seeds for the Iterated Matching Pennies. (a) Average Reward (per step) (b) CHAOS' probability of cooperation conditioned on state and (c) state visitation.

## 5.2 Baseline Comparisons

We compare our method against three baselines: a Naive Learner, M-FOS and GS. A Naive Learner (NL) does not account for the learning of the co-player. It updates at the end of each inner-episode with learning rate $\alpha$:

$$\phi_i^{t+1} = \phi_i^t + \alpha \nabla_{\phi_i^t} \mathcal{R}^i(\phi_i^t, \phi_{-i}^t) \tag{2}$$

In the IPD, our NL is parameterised as a tabular policy trained using PPO (Schulman et al., 2017). In the Coin Game, the NL is parameterised by a recurrent neural network trained using PPO. The specific implementation details are provided in Appendix D.

For M-FOS and GS, we optimise both methods using Evolution Strategies (Salimans et al., 2017), in line with the original implementations (Lu et al., 2022; Balaguer et al., 2022). For M-FOS, we use the hierarchical architecture used in its Coin Game results since we are using neural networks for these environments instead of simple tabular policies. The implementation details for M-FOS and GS are provided in Appendix D and Appendix D

In every game, CHAOS is parameterised as a recurrent neural network and is trained using Evolution Strategies (Salimans et al., 2017). We used the Jax library (Bradbury et al., 2018) with the Haiku framework (Hennigan et al., 2020) to implement our neural networks. For the Evolution strategies, we relied on the Evosax library (Lange, 2022). Our experiments were performed on NVIDIA A40 and V100 GPUs. Adddditional implementation details and hyperparameters for each game are provided in Appendix D. Furthermore, the whole codebase will be released upon acceptance.

## 5.3 Ablations

To evaluate the importance of *context* and *history* we apply the following ablations.

**The Hardstop Challenge** during a trial, after $k$ episodes the co-player no longer takes learner updates. In the situation where the co-player no longer updates, optimal behaviour would be to exploit this fixed policy (effectively stop shaping). We choose $k = 2$ to be less than the number of episodes required for a shaper to reach to ZD- Extortion-like policies. This challenge tests if shapers' can 1) identify the sudden change in a co-player's learning dynamics 2) react and deploy a more suitable exploitative policy. We evaluate CHAOS against GS, to compare context and context-less shaping methods.

Figure 4: Illustration of a special state of the Coin Game where memory-less agents can infer conventions. If both agents are on the same square, we can infer that an agent has broke convention.

**The Only-History Challenge** we reset the hidden state of CHAOS between episodes, removing its ability to use *context* to shape (CHAOS reset memory). In this challenge, shapers must infer co-player's current policy by only using the history. We evaluate the shaper within the IMP environment, over different episode lengths, to limit the relative strength of *history*.

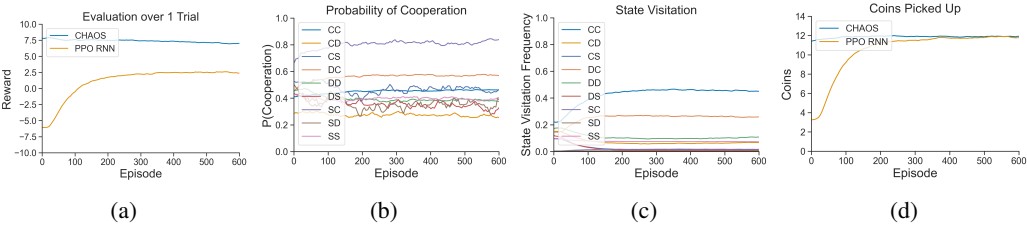

Figure 5: Evaluation results over a single trial (with novel-coplayer) compromising over 5 seeds for the Coin Game. (a) Reward, (b) the Shaper's frequency of picking up its own colour coin, (c) state visitation, and (d) the number of coins picked up per episode.

## 6  RESULTS

In this section we report and compare the results between CHAOS and our baselines on our environments. We find that CHAOS shapes achieves state-of-the-art results in the IPD. On the IMP game, CHAOS achieves state-of-the-art results, outperforming M-FOS, GS and other baselines. Next we demonstrate that CHAOS is scalable, as it achieves comparable shaping in the Coin Game. Finally through a series of ablations we demonstrate the importance of context and history for effective shaping. Similar to Meta-RL test protocol, for the evaluation we evaluate the performance of shapers against a novel POSG $\mathcal{M}$ and co-player initialisation $\phi_{-i}$.

**Iterated Prisoner's Dilemma** In the IPD, we inspect the converged reward for each shaping algorithm against a PPO agent. Here, CHAOS shapes its co-player more aggressively than the baselines (see Table 1), achieving an average return of -0.13 per episode. All shaping baselines reach ZD-extortion-like policy. CHAOS switches policies during a co-player's learning, switching from cooperation to exploitation (see Figure **??**). In Figure 2c, we display the state visitation over one trial. It shows that a fully trained CHAOS agent pursues a tit-for-tat like-strategy to encourage cooperation within the first 5 episodes before pursing an excessively exploitative policy. At this point, the co-player is shaped, as is it unable to move to another equilibrium.

**Iterated Matching Pennies** In the IMP, CHAOS exploits its opponent to achieve a score of $(0.80, -0.80)$ (see Table 1). As expected, GS cannot shape the opponent, achieving a score close to the Nash Equilibrium. Without having any context, it is not possible to shape the opponent because the opponent can also switch to a random strategy to at least achieve a score of 0. Thus dynamic-shaping, is required to find exploitative strategies. We find that M-FOS another dynamic shaping method is able to find exploitation too.

**Coin Game** CHAOS outperforms M-FOS in Table 1, providing evidence that CHAOS is scalable to more complex policies. CHAOS demonstrate shaping, as indicated by the payoffs of the shaper vs the co-player. To verify that CHAOS has shaped (as opposed to simply obstructed the co-players learning) we also inspect player's ability to pick up coins see Figure 5d.

Cooperation in the Coin Game means that players only pick up coins of their colour. Thus evaluating % of cooperating actions is useful. We adapt our analysis from the IPD to the Coin Game. We extend the four states from the IPD (CC, CD, DC, DD) to also include the start convention S (before a player picks up a coin). At the start of an episode, the state is SS as neither player has committed cooperative or defective behaviour. The state remains SS until a player picks up a coin and breaks one of the other 7 states. For example, observing a low probability of cooperating when a shaper has been exploited (CD), demonstrates that shapers punish co-players who break cooperation.

Figure 5b demonstrates how CHAOS uses context to effectively shape its co-players. Shapers have two axes on which to evaluate their co-players, 1) how competent they are and 2) how cooperative they are. We measure competency as how many coins a player can pick up. If a co-player is not competent yet (say at the beginning of its training), we would expect a shaper to exploit this behaviour. In Figure 5e, the difference between cooperating in SC and SS highlights how CHAOS uses context to evaluate exploitability of its co-player. In SS, when both agents have not picked up coins, CHAOS probes for exploitability by not cooperating, whereas in SC, where the co-player has already demonstrated it is competent, CHAOS follows convention. Figure 5c demonstrates how CHAOS exploits its co-player. We expect efficient shapers to spend the majority of their time either

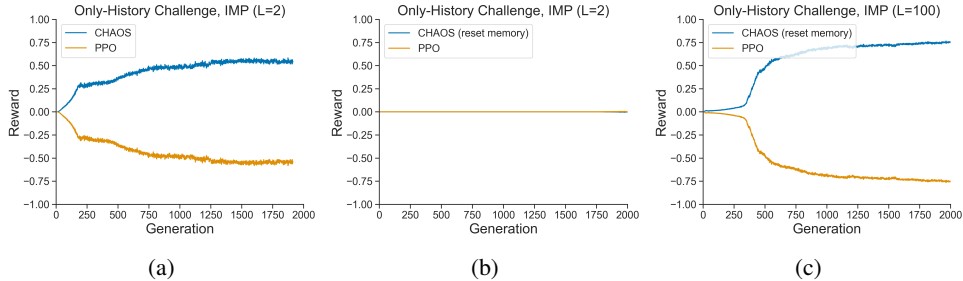

Figure 6: The Only-History Challenge: Training curves in the IMP with episode length = 2 for (a) CHAOS and (b) CHAOS without context. Note that in short time-spans, history can not be used context can enable shaping. Additionally (c) CHAOS without context in IMP with episode length = 100 shows with sufficient timespan, history can be used to shape.

cooperating (CC) or exploiting their co-players can exploit co-players (DC), here we see these two are the most likely states observed during training (50% and 20% respectively).

We found GS produces comparable results to CHAOS. At first this is surprising, since GS is a feedforward network and does not have access to the history. Therefore, in principle it should not be able to retaliate against a defecting agents since it has no memory of their past actions. However, close investigation of the problem setting shows that due to the specific environment dynamics, the *current state* is often indicative of *past actions*. For example, when the two agents are on the same square, in all likelihood one of the agents defected (see Figure 4). Similarly, if the agents are currently

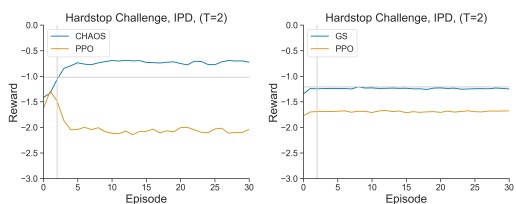

Figure 7: The Hardstop Challenge: Average reward over a single trial for CHAOS (left) and GS (right) against a naive learner in the IPD.

standing a on a coin of a given colour, this coin was picked up on the last time step. This illustrates that Coin Game allows for simple shaping strategies that do not require *context* or *history*, limiting its utility as a benchmark for investigating these aspects.

**Ablations** CHAOS outperforms GS on the hardstop challenge. CHAOS demonstrates dynamic shaping by switching strategies at $t = 2$ episodes, when the hardstop is triggered (see Fig. **??**). In contrast, GS's policy is fixed throughout a trial and thus can not exploit the hardstop.

In the Only History challenge, when playing the IMP with a small number of inner-episodes ($E = 2$), we expect shapers without history to be unable to identify co-players current learning and thus be unable to shape. We find that CHAOS is capable of this, whilst CHAOS( without history) is unable to shape agents 6. Interestingly, we also found that with a longer inner-episode length ($E = 100$), CHAOS is able to use *context* to shape its co-player 6c. This show that both context and history are required over different environments to encode co-players learning.

## 7 CONCLUSION

In this paper, we introduce CHAOS, a shaping method capturing both learning context and history, suitable for high-dimensional games. We formalise the concept of history and context for shaping and analyse their respective roles empirically. We demonstrate state-of-the-art performance on a set of iterated matrix games. We identify a fundamental problem in the widely-used Coin Game.

When multiple agents interact in a shared environment, the actions of each any one agent influence the rewards and environment faced by others, and through their learning ultimately affect their behaviour. Shaping, i.e. constructing agents that can effectively leverage this interconnection, has emerged as a sub-field in Mult-agent Reinforcement learning and has received considerable attention in recent years. CHAOS substantially expands the current capabilities of shaping agents by allowing them to react to changes to the co-players' learning dynamics as well as to predictable patterns in their within-episode behaviour, thus resulting in significantly more effective shaping.

## 8 ETHICS STATEMENT

Shaping can be for good and bad. Empirically, shaping has lead AIs to find more prosocial solutions. However, one can imagine scenarios where shaping can be used with a negative impact on society. Investigating shaping is important to prevent misuse of the paradigm. We are at the beginning of fundamental research in shaping and better understanding the necessary components to achieve shaping will help us to better control shaping agents. Opponent Shaping is still early in an early phase of development and practical implications are limited so immediate negative societal infuence is unlikely.

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

# A    MATRIX GAME DETAILS

Here we present details of training of shaping agents in Iterated Matrix Games.

## A.1    PAYOFF MATRICES

Table 2: Payoff Matrices

|     | C       | D       |
|-----|---------|---------|
| C   | (-1,-1) | (0, -3) |
| D   | (-3, 0) | (-2, -2)|

(a) Iterated Prisoners Dilemma (IPD)

|     | H       | T       |
|-----|---------|---------|
| H   | (1,-1)  | (-1, 1) |
| T   | (-1, 1) | (1, 1)  |

(b) Iterated Matching Pennies (IMP)

## A.2    TRAINING DETAILS

We present training curves for both IPD and IMP below.

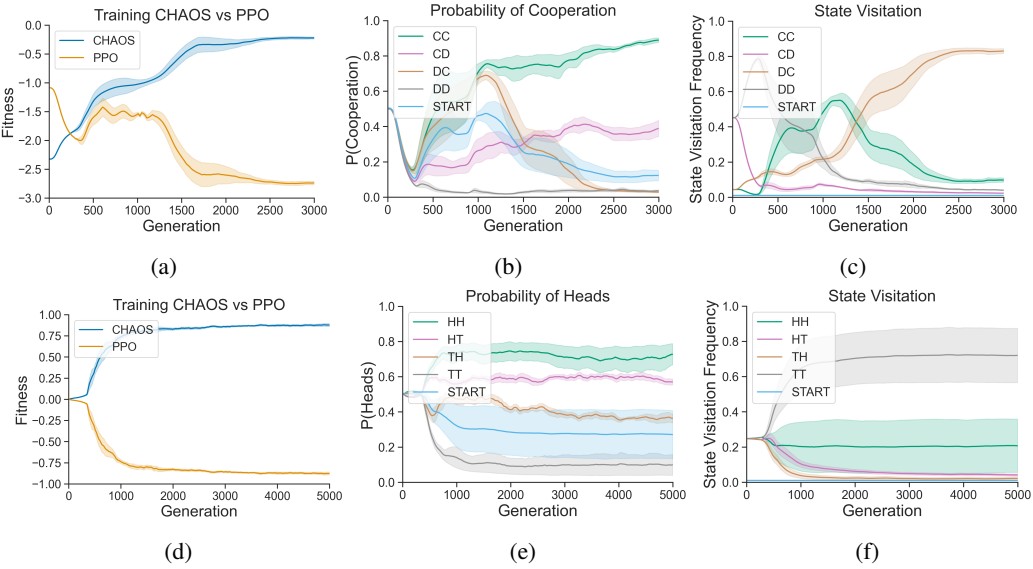

Figure 8: Training results in the finite IPD over 5 seeds for CHAOS. Here we display the (a) fitness, (b) conditional probability of cooperation, and (c) state visitation. Training results in the IMP over 5 seeds for CHAOS. (d) Fitness (e) Empirical probability of Cooperative action conditioned by state and (f) state visitation.

# B    COIN GAME DETAILS

## B.1    EXPERIMENTAL PROTOCOL

We define an experimental protocol for producing the correct conditions for shaping

**Experimental Protocol** We verify that an agent can learn against three competent opponents. The first agent is *Good* Greedy, picking up coins indiscriminately but prioritizing its own colour coin if it is equidistant from two coins; the second agent is *Evil* Greedy, picking up coins indiscriminately but prioritizing its opponent's coin if it is equidistant from two coins; the third agent is the current agent pre-trained to competency via self-play. After verifying that an agent learns competency against these three opponents, we set the parameters of the trial to reflect the time and scale required for the agent to become competent against a competent opponent.

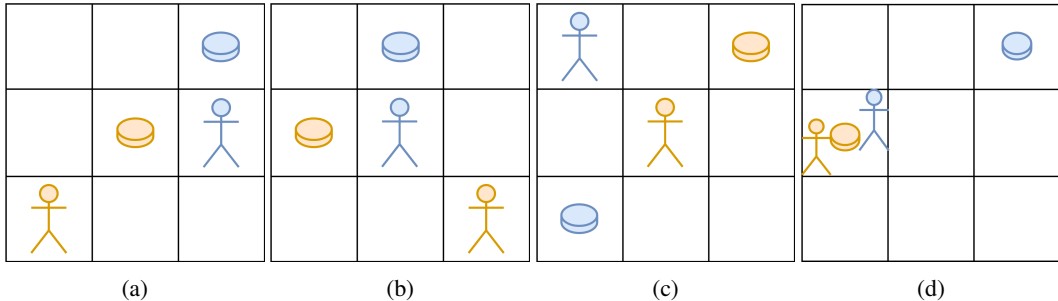

Figure 9: Illustration of (a) a state observed by both agents in the non-egocentric Coin Game and (b)-(c) the same state observed by the blue and orange agent, respectively, in the egocentric Coin Game. (d) Illustration of a special state where memory-less agents can infer conventions.

**Sanity** We verify that PPO RNN can learn against competent opponents in the Coin Game in Figure 10. PPO RNN learns to pick up the same number of coins as a pre-trained PPO RNN, Good Greedy, and Evil Greedy. These hyperparameters are the same ones used during meta-learning.

**Coin Game Adjustments**

In the Coin Game, agents struggle to learn (via reinforcement learning) when trained against a pre-trained opponent. On inspection of trajectories, we found that competent agents removed a sufficient amount of coins to restrict reinforcement learners ability to capture signal from the game.

To address this, we show that adjusting the observations such that an agent receives to an *egocentric* viewpoint (i.e. an agent always observes that it is in the centre of the grid) leads to competency against a competent opponent. In this case, we measured competency as an agent's ability to pick up coins. *Competent* agents were those who picked up a similar number of coins to those trained against a stationary agent. Throughout the rest of the paper, we refer to the original version of Coin Game as *non-egocentric* Coin Game and the modified observation version as *egocentric* Coin Game. In addition, we deviate from the original 5 by 5 version of Coin Game to a 3 by 3 version, following the setting used in (Lu et al., 2022).

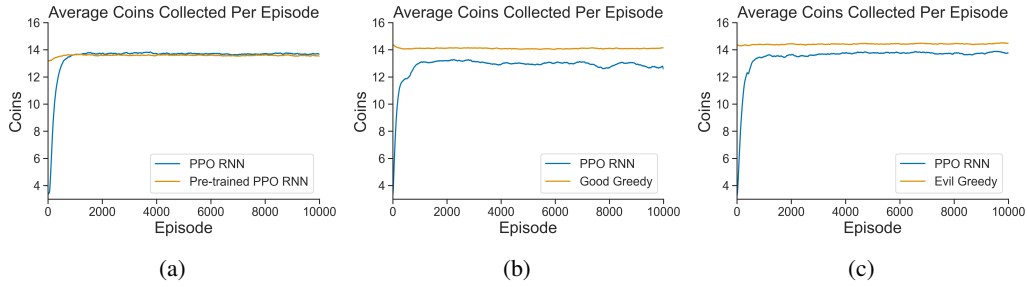

Figure 10: Results of the experimental protocol verifying that PPO RNN learns to play the egocentric Coin Game against (a) pre-trained PPO RNN (b) Good Greedy (c) Evil Greedy. Notice that the agent learns to pick up roughly the same number of coins per episode as its competent opponents.

## B.2 TRAINING DETAILS

## B.3 GENERALIZATION IN PRE-TRAINED AGENTS

## C ABLATION STUDIES

We also include additional state visitation for the hardstop challenge. This helps validate that CHAOS reacts to agents becoming exploitable after a hardstop has occurred.

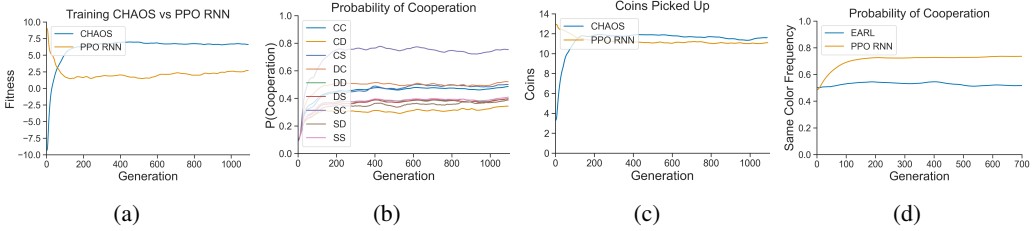

Figure 11: Training results of CHAOS vs. PPO RNN in the egocentric Coin Game. (a) Fitness, (b) the shaper's frequency of picking up its own colour coin depending on existing convention, (c) the number of coins picked up per episode, (d) both agent's frequency of picking up its own colour coin over a full episode.

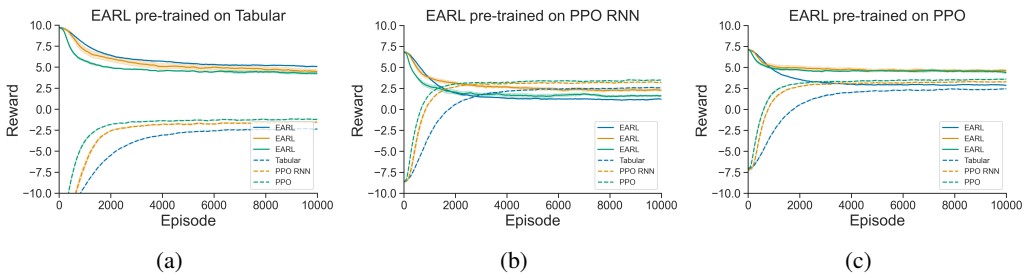

Figure 12: Results of training versus a CHAOS agent pre-trained against (a) Tabular, (b) PPO RNN, and (c) PPO. Notice that the CHAOS agent pre-trained against Tabular outperforms the CHAOS agent pre-trained against PPO RNN. We hypothesize that CHAOS finds "true" shaping against the Tabular agent but a sub-optimal shaping policy against PPO RNN, due to its complexity. Table **??** shows that CHAOS outperforms PPO RNN $(7.08, -7.36)$ by a wider margin than against Tabular $(6.67, -6.51)$, indicating that CHAOS finds a shaping policy that performs well against PPO RNN, but that this shaping policy does not generalize to other agents as robustly as the shaping policy learned against Tabular.

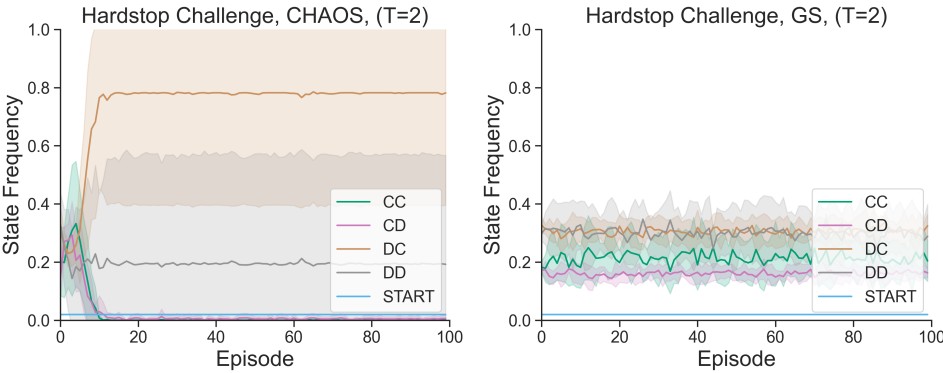

Figure 13: State visitation for the Hardstop Challenge with shapers (a) CHAOS and (b) GS. Here we see CHAOS responds to co-players frozen stationary policy by moving into either DD (the best response to a defective agent) or DC (the best response to a fully cooperative agent), whilst GS is unable to adjust its approach after the co-player stop learning, continuing to plays the sub-optimal shaping strategy)

# D  HYPER-PARAMETERS

| Hyperparameter | Value |
|---|---|
| Number of Actor Hidden Layers | 1 |
| Number of Critic Hidden Layers | 1 |
| Torso GRU Size | [25] |
| Length of Meta-Episode | 100 |
| Length of Inner Episode | 100 |
| Number of Generations | 5000 |
| Batch Size | 100 |
| Population Size | 1000 |
| OpenES sigma init | 0.04 |
| OpenES sigma decay | 0.999 |
| OpenES sigma limit | 0.01 |
| OpenES init min | 0.0 |
| OpenES init max | 0.0 |
| OpenES clip min | -1e10 |
| OpenES clip max | 1e10 |
| OpenES lrate init | 0.01 |
| OpenES lrate decay | 0.9999 |
| OpenES lrate limit | 0.001 |
| OpenES beta 1 | 0.99 |
| OpenES beta 2 | 0.999 |
| OpenES eps | 1e-8 |

Table 3: Hyperparameters for CHAOS in Iterated Prisoner's Dilemma

| Hyperparameter | Value |
|---|---|
| Number of Minibatches | 4 |
| Number of Epochs | 2 |
| Gamma | 0.96 |
| GAE Lambda | 0.95 |
| PPP clipping epsilon | 0.2 |
| Value Coefficient | 0.5 |
| Clip Value | True |
| Max Gradient Norm | 0.5 |
| Entropy Coefficient Start | 0.02 |
| Entropy Coefficient Horizon | 2000000 |
| Entropy Coefficient End | 0.001 |
| Learning rate | 1 |
| ADAM epsilon | 1e-5 |

Table 4: Hyperparameters for Tabular-PPO in Iterated Prisoner's Dilemma

| Hyperparameter | Value |
| --- | --- |
| Number of Actor Hidden Layers | 2 |
| Number of Critic Hidden Layers | 2 |
| Network Hidden Size | [16, 16] |
| Length of Meta-Episode | 100 |
| Length of Inner Episode | 100 |
| Number of Generations | 5000 |
| Batch Size | 100 |
| Population Size | 1000 |
| OpenES sigma init | 0.04 |
| OpenES sigma decay | 0.999 |
| OpenES sigma limit | 0.01 |
| OpenES init min | 0.0 |
| OpenES init max | 0.0 |
| OpenES clip min | -1e10 |
| OpenES clip max | 1e10 |
| OpenES lrate init | 0.01 |
| OpenES lrate decay | 0.9999 |
| OpenES lrate limit | 0.001 |
| OpenES beta 1 | 0.99 |
| OpenES beta 2 | 0.999 |
| OpenES eps | 1e-8 |

Table 5: Hyperparameters for GS in Iterated Prisoner's Dilemma

| Hyperparameter | Value |
| --- | --- |
| Number of Actor Hidden Layers | 1 |
| Number of Critic Hidden Layers | 1 |
| Actor GRU Hidden Size | 16 |
| Critic GRU Hidden Size | 16 |
| Meta Agent Gru Hidden Size | 16 |
| Hidden Layer Size | 16 |
| Length of Meta-Episode | 100 |
| Length of Inner Episode | 100 |
| Number of Generations | 5000 |
| Batch Size | 100 |
| Population Size | 1000 |
| OpenES sigma init | 0.04 |
| OpenES sigma decay | 0.999 |
| OpenES sigma limit | 0.01 |
| OpenES init min | 0.0 |
| OpenES init max | 0.0 |
| OpenES clip min | -1e10 |
| OpenES clip max | 1e10 |
| OpenES lrate init | 0.01 |
| OpenES lrate decay | 0.9999 |
| OpenES lrate limit | 0.001 |
| OpenES beta 1 | 0.99 |
| OpenES beta 2 | 0.999 |
| OpenES eps | 1e-8 |

Table 6: Hyperparameters for MFOS in Iterated Prisoner's Dilemma

| Hyperparameter | Value |
|---|---|
| Number of Actor Hidden Layers | 1 |
| Number of Critic Hidden Layers | 1 |
| Torso Gru Size | [16] |
| Length of Meta-Episode | 600 |
| Length of Inner Episode | 16 |
| Number of Generations | 3000 |
| Batch Size | 100 |
| Population Size | 4000 |
| OpenES sigma init | 0.04 |
| OpenES sigma decay | 0.999 |
| OpenES sigma limit | 0.01 |
| OpenES init min | 0.0 |
| OpenES init max | 0.0 |
| OpenES clip min | -1e10 |
| OpenES clip max | 1e10 |
| OpenES lrate init | 0.01 |
| OpenES lrate decay | 0.9999 |
| OpenES lrate limit | 0.001 |
| OpenES beta 1 | 0.99 |
| OpenES beta 2 | 0.999 |
| OpenES eps | 1e-8 |

Table 7: Hyperparameters for EARL in Iterated Matching Pennies

| Hyperparameter | Value |
|---|---|
| Number of Minibatches | 8 |
| Number of Epochs | 2 |
| Gamma | 0.96 |
| GAE Lambda | 0.95 |
| PPO clipping epsilon | 0.2 |
| Value Coefficient | 0.5 |
| Clip Value | True |
| Max Gradient Norm | 0.5 |
| Anneal Entropy | False |
| Entropy Coefficient Start | 0.02 |
| Entropy Coefficient Horizon | 2000000 |
| Entropy Coefficient End | 0.001 |
| LR Scheduling | False |
| Learning Rate | 0.005 |
| ADAM Epsilon | 1e-5 |
| With CNN | False |

Table 8: Hyperparameters for PPO Memory and Tabular in the Coin Game

| Hyperparameter | Value |
| --- | --- |
| Number of Actor Hidden Layers | 1 |
| Number of Critic Hidden Layers | 1 |
| Hidden Size | [16] |
| Length of Meta-Episode | 600 |
| Length of Inner Episode | 16 |
| Number of Generations | 3000 |
| Batch Size | 100 |
| Population Size | 4000 |
| OpenES sigma init | 0.04 |
| OpenES sigma decay | 0.999 |
| OpenES sigma limit | 0.01 |
| OpenES init min | 0.0 |
| OpenES init max | 0.0 |
| OpenES clip min | -1e10 |
| OpenES clip max | 1e10 |
| OpenES lrate init | 0.01 |
| OpenES lrate decay | 0.9999 |
| OpenES lrate limit | 0.001 |
| OpenES beta 1 | 0.99 |
| OpenES beta 2 | 0.999 |
| OpenES eps | 1e-8 |

Table 9: Hyperparameters for GS in Coin Game

| Hyperparameter | Value |
| --- | --- |
| Number of Actor Hidden Layers | 1 |
| Number of Critic Hidden Layers | 1 |
| Actor GRU Hidden Size | 16 |
| Critic GRU Hidden Size | 16 |
| Meta Agent Gru Hidden Size | 16 |
| Hidden Layer Size | 16 |
| Length of Meta-Episode | 100 |
| Length of Inner Episode | 100 |
| Number of Generations | 5000 |
| Batch Size | 100 |
| Population Size | 1000 |
| OpenES sigma init | 0.04 |
| OpenES sigma decay | 0.999 |
| OpenES sigma limit | 0.01 |
| OpenES init min | 0.0 |
| OpenES init max | 0.0 |
| OpenES clip min | -1e10 |
| OpenES clip max | 1e10 |
| OpenES lrate init | 0.01 |
| OpenES lrate decay | 0.9999 |
| OpenES lrate limit | 0.001 |
| OpenES beta 1 | 0.99 |
| OpenES beta 2 | 0.999 |
| OpenES eps | 1e-8 |

Table 10: Hyperparameters for MFOS in Coin Game

