# OpenReview forum: "Context and History Aware Other-Shaping"
_ICLR.cc/2023/Conference — Submitted to ICLR 2023_

### Official Review · Reviewer_PvhL · 2022-10-16

**Confidence:** 4
**Correctness:** 2
**Technical Novelty And Significance:** 2
**Empirical Novelty And Significance:** 2
**Recommendation:** 3

**Clarity, Quality, Novelty And Reproducibility:**

Clarity: poor

Quality: poor

Novelty: marginal

**Strength And Weaknesses:**

Strength:
- Some empirical results.

Weaknesses:
- The novelty is unclear.
- Presentation lacks clarity and precision. The theory background is incomplete and vague, also contains several mistakes/undefined parts.
- No actual result is presented beyond some empirical evaluation, which is neither conclusive nor sufficient.
- No formal reasoning or meaningful argument is given. I am not sure what the usefulness of this work would be at all for the ICLR audience.

**Summary Of The Paper:**

The authors propose an algorithm, called CHAOS (Context and History Aware Other-Shaping) with the goal of capturing both learning context and history.

**Summary Of The Review:**

N/A

---

> ### Author Response · Authors · 2022-11-13
> **Response to PvhL**
>
> > The novelty is unclear.
>
> We addressed the concerns for novelty in our [general response](https://openreview.net/forum?id=54F8woU8vhq&noteId=nqaF-JImu0). If the reviewer finds the time, we’d appreciate a follow-up response to address further concerns about the novelty.
>
> > Presentation lacks clarity and precision. The theory background is incomplete and vague, also contains several mistakes/undefined parts.
>
> We appreciate the reviewers' feedback. We’ve made several revisions since this rebuttal and so, to improve the paper further, we’d appreciate it if the reviewer could point us to the specific instances of mistakes and undefined parts or let us know if they have been addressed with the updated version.
>
> > No actual result is presented beyond some empirical evaluation, which is neither conclusive nor sufficient
>
> We provide results on a set of experiments that is standard in the opponent shaping literature [1, 2, 3, 4, 5]. In contrast to previous work, we provide an even more in-depth analysis of the performance on the Coin Game including insightful ablation studies. We’d appreciate it if the reviewer could elaborate how we can improve our results to make them conclusive and expand on how we can extend the results to make them sufficient.
>
> > No formal reasoning or meaningful argument is given. I am not sure what the usefulness of this work would be at all for the ICLR audience.
>
> Framing Opponent Shaping as meta-learning has shown promising results, as M-FOS is the first to find a ZD-Extortion-like strategy in the Iterated Prisoner’s Dilemma. We added more context to the Background section to improve the motivation of the problem setting we are studying. We also improved the Method section to motivate the individual components of our proposed algorithm. Concisely, this work is useful to the ICLR community as it presents new results, especially ablations, to better understand opponent shaping as meta-learning. We also propose a new method, which is simpler and more performant than the previous state-of-the-art.
>
> [1] Lu, Christopher, et al. "Model-free opponent shaping." International Conference on Machine Learning. PMLR, 2022
>
> [2] Jan Balaguer, Raphael Koster, Christopher Summerfield, and Andrea Tacchetti. The good shepherd: An oracle agent for mechanism design. arXiv preprint arXiv:2202.10135, 2022.
>
> [3] Alistair Letcher, David Balduzzi, Sebastien Racaniere, James Martens, Jakob N. Foerster, Karl Tuyls, and Thore Graepel. Differentiable game mechanics. J. Mach. Learn. Res., 20:84:1–84:40, 2019a.
>
> [4] Jakob Foerster, Richard Y Chen, Maruan Al-Shedivat, Shimon Whiteson, Pieter Abbeel, and Igor Mordatch. Learning with opponent-learning awareness. In Proceedings of the 17th International Conference on Autonomous Agents and MultiAgent Systems, pp. 122–130, 2018
>
> [5] Schafer, F. and Anandkumar, A. Competitive gradient descent. In Advances in Neural Information Processing Systems, volume 32, pp. 7623–7633, 2019.

---

### Official Review · Reviewer_Ae1h · 2022-10-21

**Confidence:** 3
**Correctness:** 2
**Technical Novelty And Significance:** 2
**Empirical Novelty And Significance:** 2
**Recommendation:** 3

**Clarity, Quality, Novelty And Reproducibility:**

- I found the paper surprisingly challenging to follow without having looked at the previous work (M-FOS).
- The novelty appears to me to be marginal, empirically demonstrating an instance of the prior work.
- The authors include detailed hyperparameter listings, but insufficient details to implement their algorithm. Without source code being released I would not expect this work to be reproducible.

**Strength And Weaknesses:**

**Strengths**
 - The authors demonstrate that memory persistent across episodes does not confound their results, which I suspected during reading the first portion of the paper.

**Weaknesses**
 - This work appears to have significant overlap with the prior work M-FOS to the point where I'm not sure if it is different. Inspection of the Algorithm block from this paper and the previous are almost identical. I believe the CHAOS one also contains a typo and that the phi on line 13 should be theta. Moreover, in Section 4 of the M-FOS paper they directly mention that genetic algorithms (a type of evolutionary strategy) would work well within this framework.
 - The method section is challenging to follow and not technical enough to be confident in how the algorithm actually works. I would encourage the authors to add exact optimization functions and how they fit within the overall algorithm.
 - If it is true that the previous work cannot handle settings where DNN implement policies, a statement I am not convinced of from this manuscript, I would like to have seen a demonstration of this algorithm in such a setting and explanation as to why the previous work could not handle it.

**Summary Of The Paper:**

This paper investigates training an agent that's own learning algorithm attempts to "shape" how its' co-players learn. This shaping ideally should cause it so that the co-players' play an equilibrium that is favorable to the agent employing the shaping. This technique was previously studied in the Model-Free Opponent Shaping paper referred to in the abstract. They show that a policy trained through evolutionary strategies is able to employ such a co-player shaping policy.

**Summary Of The Review:**

I do not think the paper as it currently exists is ready for publication. I am not convinced that this work is significantly different enough from the previous work to justify publication (it appears to just be allowing a RNN's state to persist across episode). Additionally, I found the prose rather challenging to follow and the claims and method not clear enough to contribute to the scientific discussion of the area.

---

> ### Author Response · Authors · 2022-11-13
> **Response to Ae1h (1/2)**
>
> > This work appears to have significant overlap with the prior work M-FOS to the p toint where I'm not sure if it is different.
>
> Thank you for raising this concern which we took as opportunity to clarify the differences and contributions of CHAOS in comparison to M-FOS. We addressed the novelty of our proposed method in the [general response](https://openreview.net/forum?id=54F8woU8vhq&noteId=nqaF-JImu0) and hope that it clarified the difference to M-FOS and GS.
>
> > Inspection of the Algorithm block from this paper and the previous are almost identical. I believe the CHAOS one also contains a typo and that the phi on line 13 should be theta.
>
> We thank the reviewer for spotting the typo!  We do agree that the algorithm blocks are almost (!) identical! However, note that in the M-FOS algorithm, the shaper parameters \phi_i are updated using a meta-policy parameterized by  \theta. In CHAOS we do not have a theta, since the underlying agent is also the shaper. For a more in-depth discussion of the difference, please see our general response. To make this more clear,  we’ve re-written the algorithm extensively to demonstrate its difference to M-FOS and added a clarifying paragraph demonstrating how this is different.
>
> > Moreover, in Section 4 of the M-FOS paper they directly mention that genetic algorithms (a type of evolutionary strategy) would work well within this framework.
>
> We apologise  for any confusion - we did not claim that ES is what makes CHAOS different to M-FOS. Indeed, it is true that M-FOS also uses Evolution Strategies (ES), which we write about in the Background section. However, M-FOS only uses ES for the matrix games and does not investigate using ES for the Coin Game.
>
> > The method section is challenging to follow and not technical enough to be confident in how the algorithm actually works. I would encourage the authors to add exact optimization functions and how they fit within the overall algorithm.
>
> We use the conventional way of describing such algorithms, i.e. see the RL^2 [1], M-FOS [2] and GS [3] descriptions. We define the problem setting formally and define our objective function. The optimization function, Evolution Strategies, is defined in the Background section. We clarified the connection to the accompanying algorithm in the updated version. We’d appreciate it if the reviewer could point us to missing but required technical descriptions, so we can further improve the paper.
>
> > If it is true that the previous work cannot handle settings where DNN implement policies, a statement I am not convinced of from this manuscript, I would like to have seen a demonstration of this algorithm in such a setting and explanation as to why the previous work could not handle it.
>
> We did not scale up the Hyperparameter-like architecture because we rely on the original M-FOS paper as support for our claim. Even in the original paper the authors have to use a different, more scalable architecture akin to Hierarchical RL.
>
> > I found the paper surprisingly challenging to follow without having looked at the previous work (M-FOS).
>
> We added more context to the Background section to better motivate the problem setting. Next, we clarified the differences to M-FOS and GS, as discussed in the general response.
>
> > The novelty appears to me to be marginal, empirically demonstrating an instance of the prior work.
>
> We respectfully disagree with the reviewers claim of limited novelty, addressing the concern about novelty in the[ general response](#General_Response). If the reviewer finds the time, we’d appreciate a follow-up response to address further concerns about the novelty.
>
> > The authors include detailed hyperparameter listings, but insufficient details to implement their algorithm.
>
> We added details to the method section to improve reproducibility. For example, we unified the notation of the pseudo-code and the method section. We also do provide the algorithms in our code contribution which we think should make the algorithm replicable.

---

> ### Author Response · Authors · 2022-11-13
> **Response to Ae1h (2/2)**
>
> >  I am not convinced that this work is significantly different enough from the previous work to justify publication (it appears to just be allowing a RNN's state to persist across episode).
>
> We respectfully disagree with the reviewer that the work’s contribution is purely in algorithmic performance. Our contribution is the investigation is within the role of context and history for other shaping and the environments this challenge is posed upon.
> We are indeed just allowing an RNN’s state to persist across episodes, just like the widely adapted RL^2 method for the single agent setting. However, allowing RNN’s states to persist has huge implications on its ability to shape, as we show in the ablations. For example, GS’s hidden states do not persist and fail to shape in the IMP. It is by no means trivial that a simple meta-learning architecture learns to shape and the community profits from the ablation studies provided in the paper. Previous work focused on the analysis of the final performance whereas this work focuses on analysing the necessary components to achieve shaping. [1, 3]
>
> >  Additionally, I found the prose rather challenging to follow and the claims and method not clear enough to contribute to the scientific discussion of the area.
>
> We improved the clarity of the writing, especially in the method section, and hope that the updated version addressed all of the reviewer’s concerns. If so, we ask the reviewer to consider increasing their score to reflect the incorporated feedback.
>
> [1] Yan Duan, John Schulman, Xi Chen, Peter L. Bartlett, Ilya Sutskever, and Pieter Abbeel. Rl$ˆ2$: Fast reinforcement learning via slow reinforcement learning. arXiv preprint arXiv:1611.02779, 2016.
>
> [2] Lu, Christopher, et al. "Model-free opponent shaping." International Conference on Machine Learning. PMLR, 2022
>
> [3] Jan Balaguer, Raphael Koster, Christopher Summerfield, and Andrea Tacchetti. The good shepherd: An oracle agent for mechanism design. arXiv preprint arXiv:2202.10135, 2022.

---

> > ### Comment · Reviewer_Ae1h · 2022-11-17
> > **Reply to Rebuttal**
> >
> > Thank you for taking the time to address our raised points.
> >
> > The method section has definitely improved with the focus on contrasting it with the two baseline algorithms. It may just be me, but the "collapsed into a single agent" analogy required a significant amount of eye-squinting, and could perhaps be made clearer. However, that may just be me.
> >
> > I agree with the author's response that there indeed contributions to the work that are nice to have empirically demonstrated. The main contribution I view as the "The Only-History Challenge" in the ablation section, which demonstrates advantage gain from persisting recurrent states across episode boundaries. I am happy to move my score up, but I regret that the amount of novelty doesn't warrant a full conference manuscript at this time. I believe further pursuing the role of inter- and intra-episodic memory and focusing on an exceptionally strong analysis of their respective roles in cooperation and competition could prove a fruitful avenue in the future for a lovely paper.

---

> > > ### Author Response · Authors · 2022-11-17
> > > **Response to Ae1h**
> > >
> > > We thank Reviewer Ae1h for their encouraging response and updating their score accordingly. We agree entirely that further pursuing the role of inter- and intra-episodic memory is a fruitful avenue! We would appreciate it if the reviewer could elaborate on the experiments they would like to see for an analysis sufficient for a conference-level paper.
> > >
> > > In the meantime, we will further improve the method section to address the remaining reviewer's concerns.

---

### Official Review · Reviewer_Hqnc · 2022-10-24

**Confidence:** 4
**Correctness:** 2
**Technical Novelty And Significance:** 2
**Empirical Novelty And Significance:** 2
**Recommendation:** 5

**Clarity, Quality, Novelty And Reproducibility:**

The paper lacks sufficient clarity in explaining the proposed method and overlooks several finer details that I have listed above. The proposed method implements RL2 from prior work for meta-learning the agent policy to shape the opponent in a 2-player matrix game. The quality of experimental analysis also needs to be improved and I have listed some of the current shortcomings above.


**Strength And Weaknesses:**

The paper contextualizes the prior work and outlines the proposed improvements compared to the related approaches M-FOS and GS. The following points require further correction / clarification.

1) There are some typos throughout the paper.
- Sec 3, POSG: reward function is $\mathcal{R}$ and not $\mathcal{\tau}$?
- Sec 3, POSG: single player case $\mathcal{I} = \{1\}$ or $N = \{1\}$?
- Page 4, Good Shepherd: The gradient is wrt $\phi^e_{-i}$?
- Page 4, Good Shepherd, last sentence in the paragraph is incorrectly framed?
- Sec 4, paragraph 2: Inconsistent notation in policy ($\phi$, $\theta$) and distribution ($\rho$).
- Sec 5.3: “...1)identifying the … an co-player’s …” -> “..identify…. a co-player’s…”
2) Algorithm 1: Line 9 - is the naive learner updated every time step or after T steps in the trial? There is no mention of E episodes. How are $\theta_m$ and $\theta_n$ different? Line 13 - update $\phi$ or update $\theta_m$ or $\theta_n$?
3) It would help to clearly describe the evaluation setting in the meta-learning framework. Are the results reported for 1 trial after the training has converged for both the meta-learner and the opponent? Is the opponent being trained from scratch during the evaluation trial? Is the opponent reset / re-initialized at the beginning of each trial during training as well as evaluation?
4) It would help to clearly describe the legends in subfigures showing probability of cooperation conditioned by state and the state visitation frequency. Particularly, it was unclear to me, for example in Fig 2b whether CC is the action of the (opponent, meta-learner) or (meta-learner, opponent) in the previous step? What are the corresponding states in Fig 5b / 5e / 5f ?
5) Sec 6, Coin game: “Both GS and CHAOS demonstrate shaping … (see Fig 5).” - I did not understand this, needs further explanation.
6) Sec 6, Ablations: “.. when the hardstop is triggered (see Fig 6c).” - I don’t think this is shown in Fig 6c since it contradicts the description in the caption for Fig 6c.
7) Fig 6: Is the single trial evaluated after the training has converged for both CHAOS and PPO?
8) The x-axis has a different range in Fig 6d compared to Fig 6c and 6e - it does not result in a fair comparison. The plot for ‘PPO’ should be the same in Fig 6c and Fig 6d and maybe we will see this if the range / scale of the x-axis is fixed. The paper says that in contrast with Fig 6d, Fig 6e shows with sufficient timespans, history can be used to shape in the absence of context  - but the curves are almost identical till 200 generations in Fig 6e as well as Fig 6d.
9) Why are the curves for PPO different in Fig 6a and Fig 6b? Are they comparable? Can you show the state visitation frequency with CHAOS, GS and PPO for this ablation study?


**Summary Of The Paper:**

The paper proposes a meta-learning approach to shaping the policy of other agents in a multi-agent learning setting. The proposed method, called CHAOS, meta-trains an RNN agent policy that utilizes both intra-episode history and inter-episode context information to select actions so that it can successfully shape the policies learned by the other agents to maximize its own rewards. Experiments are performed on 2-player matrix games to show that CHAOS outperforms or matches previously proposed meta-learning based other shaping algorithms that become computationally expensive with increasing action space dimensions or that do not utilize context and history information in their policy.


**Summary Of The Review:**

I recommend rejection of the paper in its current form. The description of the proposed method and the implementation and evaluation details need to be further elaborated and the experimental results should address the questions I have raised above.

---

> ### Author Response · Authors · 2022-11-13
> **Response to reviewer Hqnc (1/2)**
>
> We want to thank the reviewer for the thorough read and particularly detailed review which helped us to improve the paper! We’ve implemented all this feedback directly into the paper and summarize the changes below.
>
> > There are some typos throughout the paper.
>
> We have addressed these typos in the updated version of the paper. We appreciate your attention to detail.
>
> > Algorithm 1: Line 9 - is the naive learner updated every time step or after T steps in the trial? There is no mention of E episodes. How are θm and  θn different? Line 13 - update ϕ or update  θm  or θn ?
>
> Yes, we apologise for the confusion regarding the population index m and the naive learner index n. We removed the unnecessary indexing of the ES population to make the algorithm clearer to understand.
>
> > It would help to clearly describe the evaluation setting in the meta-learning framework. Are the results reported for 1 trial after the training has converged for both the meta-learner and the opponent? Is the opponent being trained from scratch during the evaluation trial? Is the opponent reset / re-initialized at the beginning of each trial during training as well as evaluation?
>
> Thank you for the comment, to make this clear, we’ve moved the training curves into the Appendix and made our results commentary focus specifically on the evaluation runs. The results are reported for 1 trial for both meta-learner and the opponent. The opponent is initialised randomly and we set the number of episodes to be sufficient for the opponent to learn. (We include further details of this sanity check within the Appendix B.1
>
> > It would help to clearly describe the legends in subfigures showing probability of cooperation conditioned by state and the state visitation frequency. Particularly, it was unclear to me, for example in Fig 2b whether CC is the action of the (opponent, meta-learner) or (meta-learner, opponent) in the previous step? What are the corresponding states in Fig 5b / 5e / 5f ?
>
> We apologise for the confusion this has caused. In the literature, CD, refers to Cooperation by player 1 and Defection by player 2. In particular in the shaping literature, the actions are (shaper, co-player). We’ve added a clarification to the rewards table and for Figure 5, we’ve added a clarifying paragraph explaining how we calculate states for the Coin Game.
>
> > Sec 6, Coin game: “Both GS and CHAOS demonstrate shaping … (see Fig 5).” - I did not understand this, needs further explanation.
>
> We agree with the reviewer that our explanation was lacklustre in the original submission and we improved on it in the updated submission.
>
> Just like in the IPD, successful shaping in the Coin Game involves constantly keeping the co-player between the two highest reward states, CC and DC, switching between cooperating with the co-player and exploiting them.
> We adapt our analysis from the IPD to the Coin Game. We extend the four states from the IPD (CC, CD, DC, DD) to also include the start state S (before a player picks up a coin). At the start of an episode, the state is SS as neither player has cooperated or defected. The state remains SS until a player picks up a coin and breaks into any of the other 7 states.
> In Figure 5g, CHAOS demonstrates evidence of shaping, as CC and CD are most frequently observed (50% and 20% respectively) during the evaluation trial. Another sign of shaping is if the shaper accounts for the training process of the co-player. For example, the shaper could identify if a co-player is early in its training process by observing that the co-player is incompetent at picking up coins. If a co-player is incompetent at picking up coins, they cannot punish the shaper when the shaper defects. A shaper can abuse that if he can successfully identify the training progress of the co-player.  Vice-versa, the shaper can identify who is a competent agent and then not defect on the co-player, as the competent co-player could reciprocate. In Figure 5e, the difference between cooperating in SC and SS highlights how CHAOS demonstrates this trait. In SS, when both agents have not picked up coins, CHAOS probes for exploitability by not cooperating, whereas in SC, where the co-player has already demonstrated it is competent and cooperative, CHAOS cooperates and does not probe for exploitability.
>
> > Sec 6, Ablations: “.. when the hardstop is triggered (see Fig 6c).” - I don’t think this is shown in Fig 6c since it contradicts the description in the caption for Fig 6c.
>
> This was meant to be Figure 6a. We apologise for the mix-up.
>
> > Fig 6: Is the single trial evaluated after the training has converged for both CHAOS and PPO?
>
> In Figure 6, (a,b) are a single trial (evaluation) whilst (c,d,e) are training curves over multiple trials. Note they are also in different games, we’ve made this clearer by separating the figures and adding more informative titles to each!

---

> ### Author Response · Authors · 2022-11-13
> **Response to reviewer Hqnc (2/2)**
>
> > The x-axis has a different range in Fig 6d compared to Fig 6c and 6e - it does not result in a fair comparison. The plot for ‘PPO’ should be the same in Fig 6c and Fig 6d and maybe we will see this if the range / scale of the x-axis is fixed. The paper says that in contrast with Fig 6d, Fig 6e shows with sufficient timespans, history can be used to shape in the absence of context - but the curves are almost identical till 200 generations in Fig 6e as well as Fig 6d.
>
> Figure 6c and 6d should have the same x-axis. We have updated 6c and 6e to also show this, we hope this establishes the finding more concretely.
>
> > Why are the curves for PPO different in Fig 6a and Fig 6b? Are they comparable? Can you show the state visitation frequency with CHAOS, GS and PPO for this ablation study?
>
> The curves are different because the PPO agents are learning against different agents, e.g. CHAOS or GS, which affects the learning dynamics of the game, since rewards are dependent on the actions of the co-players, i.e. see the Iterated Prisoner’s Dilemma. The fact that the PPO agents’ curves are different is evidence that the shaping methods are not equally effective at shaping. These results are comparable as both shapers start with a randomly initialised PPO agent with the same seed.
>
> > Can you show the state visitation frequency with CHAOS, GS and PPO for this ablation study?
>
> We are happy to include the footprints for the ablations of IPD and IMP. These have been added in Appendix D, Figure 13.

---

> > ### Comment · Reviewer_Hqnc · 2022-11-18
> > **Thank you for the response.**
> >
> > Thank you for updating the paper with added intuitive explanations, it has helped me understand the method better. I would be happy to increase my score.
> >
> > Although I appreciate the contribution of this paper in perhaps demonstrating a simpler approach to the problem than prior work in M-FOS, I do share some of the reservations expressed in the other reviews regarding the extent of novelty to warrant a higher score.

---

> > > ### Author Response · Authors · 2022-11-18
> > > **Response to Reviewer Hqnc**
> > >
> > > We thank Reviewer Hqnc for their response and for considering updating their score. To improve our paper further, we would like to ask what investigations would address their concerns regarding novelty. To reiterate, we propose a novel method, CHAOS, that is simpler and performs better or matches performance compared to the previous state-of-the-art. Moreover, CHAOS is formally different from M-FOS by __using only one agent__ instead of two, simplifying the problem formulation greatly. For a formal description of the differences, please see the updated Method section. Additionally, we provide novel ablation studies, investigating the roles of context and history in opponent shaping. Finally, we provide empirical evidence that the commonly-used version of the Coin Game is unsuitable for evaluating shaping for the multi-step action setting. We appreciate the reviewers time and engagement in this discussion.

---

### Official Review · Reviewer_zKsT · 2022-10-25

**Confidence:** 3
**Correctness:** 2
**Technical Novelty And Significance:** 2
**Empirical Novelty And Significance:** 2
**Recommendation:** 5

**Clarity, Quality, Novelty And Reproducibility:**

The writing for the paper is not clear, particularly in the method section (Section 4). The algorithm is not motivated and discussed well. In addition, the experiments do not well support some claims in the introduction section.

**Strength And Weaknesses:**

One of the main conerns for the paper is its novelty. The CHAOS algorithms seems to be a combination of the existing components and methods introduced in the background section. Can the author better clarity the novelty of CHAOs?

Another concern for the paper is its experimental part.
- It is claimed that the proposed method (CHAOS) is suitable for high-dimensional games. However, the most complex experimental benchmark in this paper is the Coin Game in a grid-world style. It is far less complex than the commonly used, challenging, and high-dimensional SMAC benchmark. It is therefore worth evaluating the method on more complex benchmark to better support the claim.
- It is also mentioned that "Cooperation failures, in which self-interested agents converge to collectively worst-case outcomes, are a common failure mode of MARL methods." Is this problem common for widely-used MARL algorithms like QMIX and MAPPO. In addition, how does the method compare against these popular baselines?


**Summary Of The Paper:**

This paper aims to capture co-playr learning dynamics in MARL. The authors propose Context and History Aware Other-Shaing (CHAOS) to address this problem. The CHAOS agent is a meta-learner using RNN architecture to learns to shape its co-player. The authors conduct extensive experiments on matrix games.

**Summary Of The Review:**

My main concern for the paper is its novelty and experimental evaluation. It would be better to clarify more details and motivation for proposing the algorithm in Section 4. It can also be improved by evaluating CHAOS in more commonly-used MARL benchmark such as SMAC by comparing it with recent MARL algorithms.

---

> ### Author Response · Authors · 2022-11-13
> **Response to Reviewer zKsT**
>
> We thank the reviewer for their responses and the clarifying questions! These are helpful for improving the paper and clarifying our message.
>
> > Can the author better clarify the novelty of CHAOS?
>
> We point the reviewer to the [general response](https://openreview.net/forum?id=54F8woU8vhq&noteId=nqaF-JImu0) for an in-depth discussion on the novelty of CHAOS. We also uploaded an improved version of the paper including a better explanation of the novelty of CHAOS in the method section.
>
> > Coin Game in a grid-world style. It is far less complex than the commonly used, challenging, and high-dimensional SMAC benchmark
>
> We agree that SMAC has a much richer input space than Coin Game. However Gridworlds are capable on encompassing complex behaviours see [1] and as shaping focus is upon co-players behaviour this argument is not justified. We’d also like to point out that SMAC only provides fully-cooperative games, and as such, only provides a special case of the general-sum setting that CHAOS (and previous related work) is interested in.
>
> While CHAOS can be run on SMAC, most state-of-the-art methods use Centralized-Training Decentralized-Execution, which cannot be applied to many general-sum learning settings, i.e. where sharing a value function does not make sense. This makes a meaningful comparison difficult. Furthermore, shaping does not solve any fundamental problem in fully-cooperative settings, which is why previous work does not use fully-cooperative settings in their experiments [2,3,4,5,6]. In contrast, the opponent shaping paradigm has shown promising results in general-sum learning, such as learning the famous, pro-social tit-for-tat strategy in the Iterated Prisoner’s Dilemma, where individual learners would otherwise achieve Defect-Defect.
>
> > It is therefore worth evaluating the method on more complex benchmark to better support the claim.
>
> We completely agree! For general-sum games, there are very few other high-dimensional mixed-games to evaluate this work. We are looking to extend this work to the Melting Pot [7] environment in future work. Nonetheless, we believe that CHAOS in the current state provides enough novel insights for the general-sum learning community. First, we propose a novel, simpler and more performant method in comparison to M-FOS. Second, we shine a light on the role of context and history on shaping, showcasing that context is paramount to shaping. Third, we are the first to analyse, in detail, the state-of-the-art shaping algorithms in the Coin Game, showcasing a novel evaluation protocol, including new metrics, for the Coin Game and we provide evidence that the most *widely-used* version of the Coin Game is not suitable to benchmark shaping for multi-step action environments.
>
> > Is this problem common for widely used MARL algorithms like QMIX and MAPPO?
>
> These algorithms operate within the Centralised-Training Decentralised-Execution regime, where agents use a shared-value function during training. This value function predicts the joint-rewards (as a team) for the agents being in a state, and so acts as a team optimizing a joint reward. Technically, you can understand our Coin Game experiments against a Naive Learner already as MAPO. Since the Coin Game state is fully-observable, the Naive Learner and the shaper have access to their respective “shared” value functions. Note, however, that they have different shared value functions as they have differing reward functions. QMIX was developed only for the fully-cooperative settings, so a comparison would not be very insightful. We clarified the difference to the fully-cooperative setting in the main body.
>
>
> [1] Küttler, Heinrich, et al. "The nethack learning environment." Advances in Neural Information Processing Systems 33 (2020): 7671-7684.
>
> [2] Lu, Christopher, et al. "Model-free opponent shaping." International Conference on Machine Learning. PMLR, 2022
>
> [3] Jan Balaguer, Raphael Koster, Christopher Summerfield, and Andrea Tacchetti. The good shepherd: An oracle agent for mechanism design. arXiv preprint arXiv:2202.10135, 2022.
>
> [4] Alistair Letcher, David Balduzzi, Sebastien Racaniere, James Martens, Jakob N. Foerster, Karl Tuyls, and Thore Graepel. Differentiable game mechanics. J. Mach. Learn. Res., 20:84:1–84:40, 2019a.
>
> [5] Jakob Foerster, Richard Y Chen, Maruan Al-Shedivat, Shimon Whiteson, Pieter Abbeel, and Igor Mordatch. Learning with opponent-learning awareness. In Proceedings of the 17th International Conference on Autonomous Agents and MultiAgent Systems, pp. 122–130, 2018
>
> [6] Schafer, F. and Anandkumar, A. Competitive gradient descent. In Advances in Neural Information Processing Systems, volume 32, pp. 7623–7633, 2019.
>
> [7] Leibo, J. Z., nez Guzmán, E. D., Vezhnevets, A. S., Agapiou, J. P., Sunehag, P., Koster, R., Matyas, J., Beattie, C., Mordatch, I., and Graepel, T., “Scalable Evaluation of Multi-Agent Reinforcement Learning with Melting Pot,” PMLR, 2021.

---

### Author Response · Authors · 2022-11-13
**General Rebuttal**

We want to thank the reviewers for their extensive and detailed feedback!

All of your feedback has been incorporated in an updated version of the paper that we uploaded along with this response. We address specific concerns in individual responses and show our revisions in blue. In this general response, we address common concerns across all reviewers. We identify two main concerns, which are **novelty** and **clarity**. (You could say there was CHAOS in the method section.)

First, we address **novelty**, as pointed out by R1 *“The CHAOS algorithms seems to be a combination of the existing components and methods introduced in the background section.”*,  R3 “*The novelty appears to me to be marginal, empirically demonstrating an instance of the prior work.“*, and R4 *“The novelty is unclear.”*.

To take a step back, we are investigating the problem setting of opponent shaping as meta-learning. There are two prior works, Model-Free Opponent Shaping (M-FOS) [1] and The Good Shepherd (GS) [2].

To put it concisely, CHAOS can be understood as a simpler, but empirically more performant, version of M-FOS in which the *meta-agent and underlying agent are collapsed into one agent*. Furthermore, in contrast to CHAOS, the M-FOS paper did not use Evolution Strategies to train the architecture used in the Coin Game experiments, does not report extensive results on the Coin Game and does not investigate the role of context and history on opponent shaping.

To contrast CHAOS to M-FOS in more detail, it is important to differentiate between the problem setting definition and the actual architecture.

In the problem setting of M-FOS, there are two action spaces: the meta-action space and the underlying action space. The meta-action space consists of the policy parameters of the underlying agent and the conventional action space is the action space of the game. In the CHAOS problem setting, the only action space is the action space of the underlying game. *There is only*  ***one*** *agent in CHAOS, whereas there are two agents in M-FOS.*

The CHAOS problem setting can be understood as a special case of the M-FOS problem setting where the action spaces are collapsed into one and the meta-agent is a recurrent policy.

CHAOS’ architecture is also different from the architectures proposed in M-FOS. M-FOS proposes two(!) architectures in the original paper, which are significantly different from each other. For matrix games, the meta-agent is a feedforward neural network, whose output is the exact policy parameterisation of the underlying agent, since the underlying agent’s policy can be represented by a simple table. In this case, the meta-agent is trained using Evolutionary Strategies. For the Coin Game (in which policies cannot be represented by a simple table), M-FOS proposes an architecture akin to Hierarchical RL. Both the meta-agent and the underlying agent are recurrent neural networks. The underlying agent resets their hidden state after each episode, whereas the meta-agent does not. In this architecture, the meta-agent does not output the full neural network  parameterization of the underlying agent, but instead outputs a conditioning variable, which the underlying agent uses as input. In other words, to scale up, M-FOS relies on the assumption that outputting a conditioning variable is equivalent to outputting a policy parameterisation. The conditioning variable is fixed during an episode. In contrast, CHAOS only uses one recurrent neural network, that does not reset its hidden state after an episode. *Both M-FOS and CHAOS can capture context and history, however, CHAOS only needs one agent to do so*.

Next, we compare CHAOS to GS, where the novelty can be seen much more easily. In the original GS paper, no recurrent agent is being used and there is no discussion about the notions of history or context. Thus GS cannot capture history nor context. Even though it’s not discussed, the framework is extensible to a recurrent meta-agent. However, that meta-agent would only capture history as the hidden state would be reset after each episode. By not capturing context, GS fails to shape in zero-sum games, such as the Matching Pennies, which we show in our experiments.

Next, we appreciate the reviewers' many suggestions to improve the **clarity** of the paper. We address the specific concerns in our individual responses. In general, we clarified the method section, motivating the different components to achieve shaping. We also clarified the differences between CHAOS, M-FOS and GS.

[1] Lu, Christopher, et al. "Model-free opponent shaping." International Conference on Machine Learning. PMLR, 2022
[2] Jan Balaguer, Raphael Koster, Christopher Summerfield, and Andrea Tacchetti. The good shepherd: An oracle agent for mechanism design. arXiv preprint arXiv:2202.10135, 2022.

---

### Author Response · Authors · 2022-11-17
**Kind Reminder**

Dear Reviewers,

We have significantly improved our submission thanks to your constructive comments and suggestions. Please see the revised pdf for the colour-coded revisions and the individual responses. We kindly ask the Reviewers who have yet to respond to let us know if we addressed their reviews adequately before the 18th of November. The 18th of November marks the end of discussion stage 1. After the 18th of November, discussion stage 2 begins, and we cannot update our submission anymore. If the remaining comments require revisions, we will gladly seek to address them. We thank the Reviewers for their time and consideration.

---

### Decision · Program_Chairs · 2023-01-20

**Decision:**

Reject

**Justification For Why Not Higher Score:**

The reviewers were unanimous that there is not enough novelty here.

**Justification For Why Not Lower Score:**

n/a

**Metareview: Summary, Strengths And Weaknesses:**

(a) Summary: This paper proposes an algorithm called CHAOS that uses an RNN for meta-learning, with the goal of shaping the policies used by the other agents.  The technique is evaluated on 2-player matrix games.

(b) Strengths: Empirical performance seems good on the experiments that were attempted.  One reviewer was happy that a suspicion they had while reading (that persistent memory across episodes would confound the results) was explicitly addressed in the text.  All reviewers agreed that although the initial draft had significant clarity issues, the later drafts were substantially improved.

(c) Weaknesses:  There was a strong consensus that the techniques presented did not represent a sufficiently novel contribution over existing work.  There was also a consensus that the experimental evaluation was overly limited.

**Summary Of Ac-Reviewer Meeting:**

n/a